# Real-Life Safety Profile of the 9-Valent HPV Vaccine Based on Data from the Puglia Region of Southern Italy

**DOI:** 10.3390/vaccines10030419

**Published:** 2022-03-10

**Authors:** Antonio Di Lorenzo, Paola Berardi, Andrea Martinelli, Francesco Paolo Bianchi, Silvio Tafuri, Pasquale Stefanizzi

**Affiliations:** Department of Biomedical Science and Human Oncology, Aldo Moro University of Bari, 70124 Bari, Italy; antoniodilorenzo95@gmail.com (A.D.L.); paola.berardi28@libero.it (P.B.); dott.a.martinelli@gmail.com (A.M.); frapabi@gmail.com (F.P.B.); pasquale.stefanizzi@uniba.it (P.S.)

**Keywords:** HPV, vaccines, AEFIs, causality assessment

## Abstract

Human Papillomavirus (HPV) is responsible for epithelial lesions and cancers in both males and females. The latest licensed HPV vaccine is Gardasil-9^®^, a 9-valent HPV vaccine which is effective not only against the high-risk HPV types, but also against the ones responsible for non-cancerous lesions. This report describes adverse events following Gardasil-9^®^ administration reported in Puglia, southern Italy, from January 2018 to November 2021. This is a retrospective observational study. Data about the adverse events following immunization (AEFIs) with Gardasil-9^®^ were collected from the Italian Drug Authority database. AEFIs were classified as serious or non-serious accordingly to World Health Organization guidelines, and serious ones underwent causality assessment. During the study period, 266,647 doses of 9vHPVv were administered in Puglia and 22 AEFIs were reported, with a reporting rate (RR) of 8.25 per 100,000 doses. The most reported symptoms were neurological ones (7/22). A total of 5 (22.7%) AEFIs were classified as serious, and 2 of these led to the patient’s hospitalization. In one case, permanent impairment occurred. Following causality assessment, only 2 out of 5 serious AEFIs were deemed to be consistently associated with the vaccination (RR: 0.750 per 100,000 doses). The data gathered in our study are similar to the pre-licensure evidence as far as the nature of the AEFIs is concerned. The reporting rate, though, is far lower than the ones described in clinical trials, likely due to the different approach to data collection: in our study, data were gathered via passive surveillance, while pre-marketing studies generally employ active calls for this purpose. Gardasil-9^®^’s safety profile appears to be favorable, with a low rate of serious adverse events and a risk/benefits ratio pending for the latter.

## 1. Introduction

Human Papillomaviruses (HPVs) are responsible for multiple epithelial lesions and cancers in both males and females. They are the etiological agent of cutaneous and anogenital warts which may progress to carcinoma depending on the virus’ subtype. Subtypes 16 and 18 are the ones most commonly associated with pre-cancerous and cancerous lesions of the cervix in females and of the anogenital and oropharyngeal areas in both sexes [1].

In Italy, the prevalence of high-risk HPV subtypes has been estimated at around 8% of the female population, with no significant geographical differences, while the incidence of cervical cancer is lower in the south of Italy. This uneven distribution of the incidence of HPV-related cancers has been explained by an increase in high-risk HPV prevalence in younger generations, thus predicting an increase in the burden of cervical cancer in the south of Italy in the coming decades [2].

The HPV vaccine is effective in reducing the risk of HPV infection, pre-cancerous lesions and cancer. Without HPV-vaccination, 7 out of 10 women and men will be infected with HPV, 1 in 10 women and men will get genital warts, 1 in 10 women will get pre-cancerous lesions, 1 in 100 women will develop cervical cancer and 1 in 500 women will die of cervical cancer. In men, HPV is the cause of cancer of the penis, anus, oral cavity and throat, and it has been estimated that about 1 in 200 men will get cancer caused by HPV [3,4,5]. Anti-HPV vaccination is therefore an essential means of prevention both in HPV-naïve subjects and in subjects with high-risk-HPV-related pre-cancerous lesions.

Nowadays, three vaccines against HPV are available. Gardasil^®^ (Sanofi Pasteur MSD)/Silgard^®^ (Merck Sharp & Dohme, Kenilworth, NJ, USA), a quadrivalent recombinant vaccine against the HPV types 6, 11, 16 and 18 (qHPVv), was licensed by the Food and Drug Administration (FDA) in 2006, whereas Cervarix^®^ (GlaxoSmithKline Biologicals, Brentfortd, Middlesex, UK), a bivalent recombinant vaccine for immunization against HPV types 16 and 18 (bHPVv), was licensed in 2007. Both vaccines contain non-infectious inactivated subunits, and protect against the high-risk HPV types 16 and 18, responsible for more than 70% of cervical cancer cases. The qHPVv also protects against subtypes 6 and 11, which cause most cases of genital warts [6,7].

The latest licensed HPV vaccine is Gardasil-9^®^, a 9-valent HPV vaccine (9vHPVv) which is effective against all of the HPV types covered by the qHPVv (6, 11, 16, and 18), as well as five additional oncogenic types (31, 33, 45, 52, and 58), showing strong protection against cervical infections caused by these HPV types as well as condylomas and some HPV-related cancers, including oropharyngeal, vaginal, vulvar, penile, and anal cancers [8].

The 9vHPVv was approved by the FDA on 10 December 2014, for use in females aged 9 to 26 years and males aged 9 to 15 years. For these recommendations, the Advisory Committee on Immunization Practices (ACIP) reviewed additional data on 9vHPVv in males aged 16 to 26 years. 9vHPVv and 4vHPVv are currently licensed for use in both females and males. 2vHPVv, on the other hand, is licensed for use in females only [9].

Immunization strategies in Italy are designed by the Ministry of Health and described in the National Immunization Plan (NIP). Each of the 20 Italian regions must follow the guidelines stated by the NIP, but may also offer other vaccines to target populations not covered by the National Plan itself. Furthermore, since 2012 the Ministry of Health has promoted the “Vaccination schedule for life”, an immunization schedule that follows every phase of an individual’s life with the objective of protecting them for the whole duration of their life [10].

The Italian Vaccination schedule for life, contained within the National Vaccine Prevention Plan 2017–2019, provides for two or three doses of HPV vaccine, according to the vaccine and to the patient’s age. Two doses are recommended for subjects aged from 9 to 14 years (both males and females), with a 6- to 12-month interval between the doses, while three doses are recommended for patients aged from 15 to 26 years, with administration at 0, 1 to 2, and 6 months (only for women) [10].

Puglia is a region in the south of Italy (around 4 million inhabitants); the 2018 Apulian edition of the Vaccination schedule for life follows the same immunization schedule as the national one, while also offering the HPV vaccination to adult women considered at higher risk for cervical cancer due to high-risk sexual behaviors; in this case, the HPV vaccine may be requested by the patient herself via co-payment, or offered during periodic screening for cervical pre-cancerous lesions [11].

Three years after the implementation of the universal mass-vaccination program using 9vHPVv, it is useful, from a public health perspective, to assess the real-life safety profile of this vaccine. The World Health Organization (WHO, Geneva, Switzerland) recommends surveillance of Adverse Events Following Immunization (AEFIs) during the post-marketing life of new vaccines as a mean to better understand the safety profile and effectiveness of new drugs [12]. Thanks to the revision of AEFIs’ reporting rates, post-marketing surveillance is indeed capable of detecting rare adverse events which pre-licensure studies could not observe, as well as studying the vaccine’s safety profile in subgroups that were not represented in pre-marketing trials [13,14]. The WHO has recommended the application of a standardized causality assessment methodology in order to grant a more ordinate approach to the surveillance of AEFIs, as well as to surpass the “emotional” approach, which may increase vaccine hesitancy while decreasing the quality of the surveillance data [15].

This report describes the adverse events following Gardasil-9^®^ administration reported in Puglia from 2018 to 2021, focusing on serious AEFIs and taking into consideration the causality assessment. We aim to design the product’s safety profile in a real-life scenario and compare it with the safety profile highlighted in phase-three clinical studies.

## 2. Materials and Methods

This is a retrospective observational study. Data were collected from the list of AEFIs recorded following 9vHPVv (Gardasil-9^®^) administration from January 2018 to November 2021, which was obtained from the Italian Drug Authority (AIFA, Rome, Italy) database. Reporting AEFIs is indeed mandatory for all healthcare workers in Italy, and reports must be submitted to the National Pharmacovigilance Network (RNF), an online platform managed by AIFA itself. AEFIs may also be reported by the person experiencing them or by their legal representative. The overall number of Gardasil-9^®^ doses administered during the study period in Puglia was extrapolated from the regional online immunization database (GIAVA).

For every subject who suffered from one or more AEFIs, a form was completed which included information about date of birth, gender, date of vaccine administration, and other vaccines administered at the same time. AEFIs were described by providing the following data: date of onset and date of computing in RNF, clinical characteristics, duration, treatment, final outcome, hospitalization or emergency room access, and a description of the case.

An Excel spreadsheet was used to build the database and perform the required analyses. The total reporting rate was calculated as the total number of reported AEFIs divided by the number of Gardasil-9^®^ administrations during the study period, while the annual reporting rate was calculated as the number of AEFIs that occurred in a year divided by the number of doses administered in the same year.

AEFIs were classified as “serious” or “non-serious” following WHO guidelines, that define an AEFI as serious if it results in death, is life threatening, requires in-patient hospitalization or prolongation of existing hospitalization, results in persistent or significant disability/incapacity, results in a congenital anomaly/birth defect, or requires intervention to prevent permanent damage or impairment. Additionally, in 2016, AIFA published a list of particular health conditions that must be considered as serious AEFIs when occurring after vaccination. This list is the Italian version of the European Medicines Agency’s important medical events list [16,17].

For serious AEFIs, the WHO’s causality assessment algorithm was applied in order to classify AEFIs as having a “consistent causal association”, having an “inconsistent causal association”, “indeterminate”, or “non-classifiable”. In particular, for AEFIs requiring hospitalization, the patient’s medical records were examined for a better understanding of the event’s characteristics [18]. Causality assessment was carried out by two different physicians with expertise in vaccinology and results were compared; in cases of divergent results, the literature was reviewed and a third physician was consulted in order to decide how to classify the adverse event.

## 3. Results

A total of 266,647 doses of 9vHPVv (Gardasil-9^®^) were administered in Puglia from January 2018 to November 2021. During the same period, 22 adverse events following Gardasil-9^®^ administration were reported in Puglia (reporting rate (RR): 8.25 per 100,000 doses). Reporting rates were higher during the first two years after the vaccine’s authorization, significantly decreasing over the following years (Table 1).

The overall male/female ratio of AEFIs was 0.833 (10 males vs. 12 females). The majority of AEFIs were reported in subjects aged from 10 to 18, with 15 reports out of 22 (68.2%) for subjects between 11 and 12 years of age. In detail, 17 AEFIs (77.3%) were reported in subjects aged from 10 to 14, an AEFI (4.50%) was reported for a 17-year-old subject and the remaining 4 AEFIs (18.2%) were observed in subjects over 25 years of age.

Table 2 describes the prevalence of specific signs and symptoms reported in the AEFI data. Neurological symptoms were the most common, followed by local events of pain, tenderness, oedema and/or swelling and allergic reactions.

Out of 22 AEFIs, 5 (22.7%) were classified as serious and 17 (77.3%) as non-serious, according to the latest WHO guidelines. A total of 2 out of 5 (40.0%) serious AEFIs led to the patients’ hospitalization, and one of them (20.0%) caused impairment. The RR for serious AEFIs was 1.87 per 100,000 doses.

Out of the five serious AEFIs, two (40.0%) were deemed to be consistently associated with the vaccine’s administration, while for another two of them (40.0%), no consistent causal association was found between the adverse event and the vaccination. For the fifth adverse event, the causality assessment outcome was undetermined.

The results show that 1 of the 2 AEFIs with consistent causal association caused the patient’s hospitalization, but in both cases the subjects had fully healed by the time the report was completed. The RR for vaccine-related serious AEFIs was 0.750 per 100,000 doses.

The final outcome for 15 out of 22 AEFIs (68.2%) was the patient’s complete recovery, while for three of them (13.6%) only partial improvement occurred. In total, 2 out of 22 patients (9.10%) had still not healed from the reported adverse events, while for another 2 (9.10%) the AEFI’s outcome was not known.

We will now focus on the five serious AEFIs. We will describe these adverse events based on the data provided by the reporting subjects, and causality assessment will be taken into consideration in order to better understand the outcomes of the AEFIs’ evaluations.

### 3.1. AEFI 1

The first case was reported in a female subject, aged 33 at the time of onset of the adverse event. The subject was already known to have had mild allergic reactions to food allergens. About six hours after Gardasil-9^®^ administration, the patient manifested a skin rash localized to the chest and abdomen, followed by glottis oedema on the next day. The subject was therefore hospitalized, and the adverse event was treated by intravenous corticosteroid infusion and intramuscular injection of an antihistaminic (active ingredients were not reported), which were gradually discontinued over the following days. The adverse event healed completely. The reaction was deemed to be consistently associated with the vaccine’s administration, as anaphylaxis has been observed as a rare serious adverse event following vaccination with Gardasil-9^®^.

### 3.2. AEFI 2

The second case was reported in a female subject, aged 11 at the time of onset of the adverse event. The subject was administered with Gardasil-9^®^, injected in the left deltoid, and anti-meningococcal serotype ACW_135_Y conjugated vaccine Menveo^®^, injected in the right deltoid. On the following day, the patient reported a syncopal episode with mild hypotension and significant oedema and hematoma were observed on the right arm. The subject required neither hospitalization nor pharmacological therapy, and the AEFI healed completely. The reaction was deemed to be non-associated with the vaccine’s administration, as it happened more than 12 h after the vaccine’s administration.

### 3.3. AEFI 3

The third case was reported in a male subject, aged 11 years at the time of onset of the adverse event. About one hour after Gardasil-9^®^ administration, the subject lost consciousness for approximately one minute, falling and reporting mild cranial trauma. Witnesses reported that the patient appeared pale and sweating shortly before the event. The subject was put in the Trendelenburg position and an ice-bag was placed on his head, leading to complete recovery. Hospitalization was not needed, as the symptoms resolved in a few hours. The reaction was deemed to be consistently associated with the vaccine’s administration.

### 3.4. AEFI 4

The fourth case was reported in a female subject, aged 25 at the time of onset of the adverse event. Ten days after Gardasil-9^®^ administration, the subject reported prolonged asthenia of the lower limbs, muscle stiffness, fatigue and persistent paresthesia, with such intensity that ordinary activities were impeded. Myorelaxants were administered, followed by duloxetine, and the patient was diagnosed with fibromyalgia. It is interesting to note that this AEFI was reported only three years after its occurrence, mainly due to the symptoms’ persistence and their significant impact on the patient’s quality of life. The reaction was deemed to be non-associated with the vaccine’s administration.

### 3.5. AEFI 5

The fifth case was reported in a female subject, aged 11 at the time of onset of the adverse event. The subject was administered with both Gardasil-9^®^ and anti-meningococcal serotype B recombinant absorbed vaccine Trumenba^®^ during the same immunization session. Following vaccination, the patient manifested headache, vertigo and extrasystoles with bigeminal rhythm (time of onset after vaccination was not noted in the report), and was therefore hospitalized. In hospital, laboratory analysis showed that their troponin level was 3 pg/mL, while the creatinine-kinase level was 106 U/L. No further action was taken, and the subject’s health condition improved. The adverse event was defined as undetermined following causality assessment, due to the co-administration of two different vaccines and the lack of more specific information.

## 4. Discussion

Our study describes data referring to the safety profile of Gardasil-9^®^, which is currently the most used anti-HPV vaccine. Gardasil-9^®^ is offered in Puglia to subjects aged 9 and older as part of the region’s routine vaccination schedule. The vaccine was offered actively and free-of-charge, and 266,647 vaccine doses were administered in Puglia from 2018 to 2021.

Data from the passive surveillance of AEFIs showed that more than 8 subjects out of 100,000 receiving Gardasil-9^®^ suffered from one or more adverse events. Serious AEFIs were reported in five patients, or less than 2 cases out of 100,000 administered doses, and the most common adverse events were neurological symptoms, reported in seven patients, local events of pain, tenderness, oedema or swelling, reported in six patients, and allergic reactions, reported in six patients.

Following causality assessment, a significant causal association between the adverse event and the vaccination was found for 2 out of 5 serious AEFIs. One of them was a case of loss of consciousness which occurred one hour after the vaccine’s administration, and which spontaneously resolved in a few seconds; other symptoms, such as pallor and sweating, disappeared in less than a day. The episode did not determine any permanent damage to the patient. The second AEFI which was deemed related to the vaccine was an allergic reaction with mild glottis oedema and erythema located on the chest and abdomen, which occurred in a subject with a previous history of anaphylaxis, and which was successfully treated with corticosteroid and antihistaminic drugs after hospitalization.

Permanent impairment was reported in one case, but causality assessment ruled out the hypothesis of a causal correlation between the disability and the vaccine’s administration; in this case, the patient reported asthenia, muscular stiffness, fatigue to the lower limbs and the formication of both hands and feet ten days after the vaccination, and was later diagnosed with fibromyalgia. While muscular symptoms are a common side effect of Gardasil-9^®^ [19], no data exist regarding a plausible biological association between this vaccine and fibromyalgia, which was therefore defined as a non-vaccine-related AEFI. No cases of death were reported.

Comparing these data with pre-licensure evidence, reporting rates in our study are significantly lower. The United States Food and Drug Administration (FDA), for instance, reported local events of pain, swelling and erythema, and headache in more than 10% of patients treated with Gardasil-9^®^ [19]. This discrepancy is likely due to the different surveillance methods employed in these studies: whereas pre-licensure evidence is gathered via active call, our data were collected through a passive surveillance network. Passive surveillance is in fact undermined by the risk of under-reporting, as Italian patients and healthcare professionals tend not to report adverse events, especially when mild and self-limiting. This phenomenon has already been documented by other studies of our research group [18].

On the other hand, our data are similar to pre-licensure evidence as far the symptoms’ distribution is concerned. In fact, neurological symptoms and local reactions were the most reported symptoms both in FDA authorization studies and in ours [19].

A 2018 review on HPV vaccines’ safety profile highlighted that injection-site reactions are fairly common with 9vHPVv, likely due to the greater amount of adjuvant contained in this product. Headache too was reported commonly, thus confirming the trend that emerged in our study. The higher reporting rates in this review are likely related to the active surveillance protocol employed in many of the considered studies. The review also focused on various adverse events of special interest, such as allergic reactions and anaphylaxis; the latter’s reporting rate was 0.17 per 100,000 doses in the United States, about 0.30 per 100,000 doses in Australia and Canada and 1 per 100,000 doses in the United Kingdom. These data are slightly lower than the ones regarding allergic reactions we extrapolated from the RNF, which indicate an RR of 2.25 per 100,000 doses. Therefore, despite being flawed by under-reporting, Italian data may be considered a good estimate of the real incidence rate of AEFIs as far as serious reactions are concerned [20].

A 2017 review of clinical trials and case series regarding AEFIs after HPV vaccination included a clinical trial in which systemic adverse events were reported by 59.7% of patients who were administered with 9vHPVv. However, the difference between this group and the control one was not significant. Another study included in this review mentioned significant differences between 9vHPVv and 4vHPVv, with the 9-valent vaccine causing severe injection-site reactions and systemic adverse events more frequently than the 4-valent product. Systemic AEFIs were especially frequent, having been reported in nearly 30% of patients [21].

Our study’s main strength is the high numerosity of the reference population: The Apulian population is over 4 million, and 266,647 doses of Gardasil-9^®^ were administered over the course of four years. This is a significantly larger population than the ones examined in pre-licensure clinical trials. Moreover, Gardasil-9^®^ has been licensed for use in Italy since 2018. During the immediate post-licensure period, the attention on new vaccines is higher from both the physicians’ and the patients’ perspectives. Studies that focus on the first stage of a drug’s post-marketing life are fundamental not only to identify rare AEFIs and to better understand these new products’ safety profile, but also to discern adverse events which are related to the vaccine from those that are not, building the public’s trust towards vaccinations. In addition to this, our study focuses on causality assessment for serious AEFIs, an aspect that is often overlooked by post-marketing studies [15].

On the other hand, as already stated, our study is affected by our data collection method: passive surveillance carries a high risk of under-reporting and tends to alter the serious/non-serious AEFIs ratio. One of the reported adverse events, moreover, had not undergone causality assessment, thus reducing the available information regarding serious AEFIs.

The safety profile of vaccines is currently one of the main points of argument with anti-vaccination groups. This is also one of the most important reasons for vaccination skepticism among the general public, thus representing an essential building block of effective medical information [22]. As already demonstrated by the recent withdrawal of the AstraZeneca ChAdOx-1S anti-SARS-CoV-2 vaccine from the market, failures in communication between the scientific community and the public can cause a significant distrust of the general population towards vaccination practices [23].

As far as HPV vaccines are concerned, gaining optimal vaccination coverage is especially important. According to a 2020 modelling study, the recent crisis of HPV vaccinations in Japan is expected to result in 24,600–27,300 cases of cervical cancer in 1994–2007 birth cohorts, predicting at least 5000 deaths in the next few years; a further increase in HPV-related cancers might occur if vaccination coverages are not restored in the immediate future [24].

## 5. Conclusions

The risk of AEFIs is conclusively very low in subjects both under and over 18 years of age (<0.1‰ of administered doses), reinforcing the available evidence about the favorable risks/benefits ratio for Gardasil-9^®^ [25]. Since the beginning of Gardasil-9^®^’s marketing, only one of the reported AEFIs led to permanent impairment, and it was not deemed to be consistently associated with the vaccine’s administration. Furthermore, the outcomes of all the remaining serious AEFIs were at least a partial recovery.

Effective communication between the scientific world and the public is therefore imperative in order to ensure that people keep trusting vaccination practices and understand their importance.

## Figures and Tables

**Table 1 vaccines-10-00419-t001:** AEFIs reporting rates during 2018–2021.

Year	Total Number of Administered Doses	AEFIs	RR (/100,000 Doses)
2018	68,756	5	7.27
2019	78,895	10	12.7
2020	56,411	3	5.32
2021	62,585	4	6.39

**Table 2 vaccines-10-00419-t002:** Prevalence of specific signs and symptoms described in the AEFI reports, and reporting rates (RR) per 100,000 doses.

Signs/Symptoms	N°	% out of 22 AEFIs	RR per 100,000 Doses
Neurological symptoms	7	31.8	2.62
Local pain/tenderness/oedema/swelling	6	27.3	2.25
Allergic reactions	6	27.3	2.25
Gastro-intestinal symptoms	3	13.6	1.12
Fever/hyperpyrexia/chills	2	9.09	0.750
Other symptoms	14	63.6	5.25

## Data Availability

Data regarding the total number of Gardasil-9^®^ doses administered in Puglia during the study period were extrapolated from the regional online immunization database (GIAVA). Data regarding adverse events following immunization with Gardasil-9^®^ were gathered from the Italian Drug Authority (AIFA) database.

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
