# Peer review of "Real-Life Safety Profile of the 9-Valent HPV Vaccine Based on Data from the Puglia Region of Southern Italy"

_vaccines, 2022, doi:10.3390/vaccines10030419_

Round 1

Reviewer 1 Report

Lorenzo et al. have studied the adverse events following immunization (AEFIs) with Gardasil-9 in Puglia, South of Italy, from January 2018 to November 2021. During the study period, 266,647 doses of the 9vHPV vaccine were administered and 22 AEFIs were reported (8.25 per 100,000 doses). Five cases (22.7%) were classified as serious, and 2 of these led to the patient’s hospitalization. Only 2 out of 5 serious AEFIs were deemed consistently associated with the vaccination (0.750 per 100,000 doses). In conclusion, the 9vHPV vaccine safety profile appears to be favourable, with a low rate of serious adverse events and a risk/benefits ratio pending for the latter.

The claims are properly placed in the context of the previous literature. The experimental data support the claims. The manuscript is written clearly enough that most of it is understandable to non-specialists. The authors have provided adequate proof for their claims, without overselling them. The authors have treated the previous literature fairly. The paper offers enough details of methodology so that the experiments could be reproduced.

Minor revision

The following paragraphs in the conclusion should be moved to the discussion:

"The safety profile of vaccines is currently one of the main points of argument with anti-vaccination groups. It is also one of the most important reasons for vaccination skepticism among the general public, thus representing an essential building block of correct medical information [20]. As already demonstrated by the recent withdrawal of AstraZeneca ChAdOx-1S anti-SARS-CoV-2 from the market, failures in communication between the scientific community and the public can cause a significant distrust of the general population towards vaccination practices [21]."

and

"As far as HPV vaccines are concerned, gaining optimal vaccination coverage is especially important. According to a 2020 modelling study, the recent crisis of HPV vaccinations in Japan is expected to result in 24,600-27,300 cases of cervical cancer in 1994-2007 birth cohorts, predicting at least 5,000 deaths in the next few years; a further increase in HPV-related cancers might occur if vaccination coverages are not restored in the immediate future [22]."

Add:

The HPV vaccine reduces the risk of HPV infection, precancerous lesions and cancer. Without HPV-vaccination, 7 out of 10 women and men will be infected with HPV, 1 in 10 women and men will get genital warts, 1 in 10 women will get precancerous lesions, 1 in 100 women will develop cervical cancer and 1 in 500 women will die of cervical cancer, even though Italy has a national screening program. In women, HPV is also a cause of cancer of the external genitalia of women (vulva), internal genitalia of women (vagina), anus, oral cavity and pharynx. In men, HPV is the cause of cancer of the penis, anus, oral cavity and throat. About 1 in 200 men get cancer caused by HPV.

References

Kjaer SK, Tran TN, Sparen P, Tryggvadottir L, Munk C, Dasbach E, Liaw KL, Nygård J, Nygård M. The burden of genital warts: a study of nearly 70,000 women from the general female population in the 4 Nordic countries. J Infect Dis. 2007 Nov 15;196(10):1447-54. doi: 10.1086/522863. Epub 2007 Oct 31. PMID: 18008222.

Orumaa M, Kjaer SK, Dehlendorff C, Munk C, Olsen AO, Hansen BT, Campbell S, Nygård M. The impact of HPV multi-cohort vaccination: Real-world evidence of faster control of HPV-related morbidity. Vaccine. 2020 Feb 5;38(6):1345-1351. doi: 10.1016/j.vaccine.2019.12.016. Epub 2020 Jan 6. PMID: 31917039.

Drolet M, Bénard É, Pérez N, Brisson M; HPV Vaccination Impact Study Group. Population-level impact and herd effects following the introduction of human papillomavirus vaccination programmes: updated systematic review and meta-analysis. Lancet. 2019 Aug 10;394(10197):497-509. doi: 10.1016/S0140-6736(19)30298-3. Epub 2019 Jun 26. PMID: 31255301; PMCID: PMC7316527.

Author Response

Q1. Lorenzo et al. have studied the adverse events following immunization (AEFIs) with Gardasil-9 in Puglia, South of Italy, from January 2018 to November 2021. During the study period, 266,647 doses of the 9vHPV vaccine were administered and 22 AEFIs were reported (8.25 per 100,000 doses). Five cases (22.7%) were classified as serious, and 2 of these led to the patient’s hospitalization. Only 2 out of 5 serious AEFIs were deemed consistently associated with the vaccination (0.750 per 100,000 doses). In conclusion, the 9vHPV vaccine safety profile appears to be favourable, with a low rate of serious adverse events and a risk/benefits ratio pending for the latter.

The claims are properly placed in the context of the previous literature. The experimental data support the claims. The manuscript is written clearly enough that most of it is understandable to non-specialists. The authors have provided adequate proof for their claims, without overselling them. The authors have treated the previous literature fairly. The paper offers enough details of methodology so that the experiments could be reproduced.

A1. Thank you.

Minor revision

Q2. The following paragraphs in the conclusion should be moved to the discussion:

"The safety profile of vaccines is currently one of the main points of argument with anti-vaccination groups. It is also one of the most important reasons for vaccination skepticism among the general public, thus representing an essential building block of correct medical information [20]. As already demonstrated by the recent withdrawal of AstraZeneca ChAdOx-1S anti-SARS-CoV-2 from the market, failures in communication between the scientific community and the public can cause a significant distrust of the general population towards vaccination practices [21]."

and

"As far as HPV vaccines are concerned, gaining optimal vaccination coverage is especially important. According to a 2020 modelling study, the recent crisis of HPV vaccinations in Japan is expected to result in 24,600-27,300 cases of cervical cancer in 1994-2007 birth cohorts, predicting at least 5,000 deaths in the next few years; a further increase in HPV-related cancers might occur if vaccination coverages are not restored in the immediate future [22]."

A2. Ok.

Q3. Add:

The HPV vaccine reduces the risk of HPV infection, precancerous lesions and cancer. Without HPV-vaccination, 7 out of 10 women and men will be infected with HPV, 1 in 10 women and men will get genital warts, 1 in 10 women will get precancerous lesions, 1 in 100 women will develop cervical cancer and 1 in 500 women will die of cervical cancer, even though Italy has a national screening program. In women, HPV is also a cause of cancer of the external genitalia of women (vulva), internal genitalia of women (vagina), anus, oral cavity and pharynx. In men, HPV is the cause of cancer of the penis, anus, oral cavity and throat. About 1 in 200 men get cancer caused by HPV.

References

Kjaer SK, Tran TN, Sparen P, Tryggvadottir L, Munk C, Dasbach E, Liaw KL, Nygård J, Nygård M. The burden of genital warts: a study of nearly 70,000 women from the general female population in the 4 Nordic countries. J Infect Dis. 2007 Nov 15;196(10):1447-54. doi: 10.1086/522863. Epub 2007 Oct 31. PMID: 18008222.

Orumaa M, Kjaer SK, Dehlendorff C, Munk C, Olsen AO, Hansen BT, Campbell S, Nygård M. The impact of HPV multi-cohort vaccination: Real-world evidence of faster control of HPV-related morbidity. Vaccine. 2020 Feb 5;38(6):1345-1351. doi: 10.1016/j.vaccine.2019.12.016. Epub 2020 Jan 6. PMID: 31917039.

Drolet M, Bénard É, Pérez N, Brisson M; HPV Vaccination Impact Study Group. Population-level impact and herd effects following the introduction of human papillomavirus vaccination programmes: updated systematic review and meta-analysis. Lancet. 2019 Aug 10;394(10197):497-509. doi: 10.1016/S0140-6736(19)30298-3. Epub 2019 Jun 26. PMID: 31255301; PMCID: PMC7316527.

A3. Thank you for your input, we found the suggested paragraph useful and added it to the Introduction. Minor changes were made in order to avoid redundancy and References were updated.

Reviewer 2 Report

The manuscript does not contain Line numbering which makes it very difficult to indicate locations of needed revisions.

There are many paragraphs throughout the manuscript contain only one sentence which is very poor writing style. [examples as follows] Please eliminate these 1-sentence paragraphs by combining them with the previous paragraphs or adding more information to construct a multi-sentence paragraph. The first sentence or topic sentence of a new paragraph should not use a weak indirect statement.

Introduction

3rd paragraph (p) - Anti-HPV vaccination...

10th p - Three years..

12th p - In order to..

13th p - This report..

Provide a final paragraph of the Introduction to list study objectives.

Some topic (first) sentences in paragraphs contain indirect statements (beginning with prepositions or dependent clauses) which do not provide strong introductions for new paragraphs. These should be made into direct statements by putting the subject of the sentence first.

Materials and Methods - more 1-sentence paragraphs

3rd p. - The overall...

5th p. - An Excel spreadsheet..

More information should be provided to indicate summaries of patient histories (breakdown of age, sex, race, preexisting conditions, other ailments, etc.) which may be provided by a reference to the database sources.

Results

2nd p. - The overall...

3rd p. - Table 2 (give more details on table contents and significance)

5th p. - The RR...

More detailed descriptions are needed to provide additional comparisons and details of data within both tables.

Discussion

The Discussion section should contain a comparison of study results to those of other studies on Gardasil-9 vaccine in the literature.

Conclusions

The last paragraph should indicate that more "effective" communications are needed to convince the public to use the Gardasil-9 vaccine, not necessarily that the communications need to be more "accurate". This seems to be the wrong adjective or descriptive word to use (to describe the inadequate public communications messaging).

Author Response

Q1. The manuscript does not contain Line numbering which makes it very difficult to indicate locations of needed revisions.

A1. We apologize for this inconvenience, that is related to the editorial guidelines.

Q2. There are many paragraphs throughout the manuscript contain only one sentence which is very poor writing style. [examples as follows] Please eliminate these 1-sentence paragraphs by combining them with the previous paragraphs or adding more information to construct a multi-sentence paragraph. The first sentence or topic sentence of a new paragraph should not use a weak indirect statement.

Introduction

3rd paragraph (p) - Anti-HPV vaccination...

10th p - Three years..

12th p - In order to..

13th p - This report..

Provide a final paragraph of the Introduction to list study objectives.

Some topic (first) sentences in paragraphs contain indirect statements (beginning with prepositions or dependent clauses) which do not provide strong introductions for new paragraphs. These should be made into direct statements by putting the subject of the sentence first.

Materials and Methods - more 1-sentence paragraphs

3rd p. - The overall...

5th p. - An Excel spreadsheet..

More information should be provided to indicate summaries of patient histories (breakdown of age, sex, race, preexisting conditions, other ailments, etc.) which may be provided by a reference to the database sources.

Results

2nd p. - The overall...

3rd p. - Table 2 (give more details on table contents and significance)

5th p. - The RR...

More detailed descriptions are needed to provide additional comparisons and details of data within both tables.

A2. Thank you for your remarks. We improved the text’s style as required, reducing one-sentence paragraphs to a minimum and changing indirect statements into direct sentences where possible. A final paragraph was added to the Introduction listing study objectives.

Information about our patients’ age and sex was listed in the Results section, while other information such as race and preexisting conditions was not available since it is not required by Italian RNF network when filing adverse event reports.

Q3. Discussion. The Discussion section should contain a comparison of study results to those of other studies on Gardasil-9 vaccine in the literature.

A3. The Discussion section already provides a comparison of our study results with other reviews about Gardasil-9®’s safety profile  (see Ref 20 and 21).

Q4. Conclusions. The last paragraph should indicate that more "effective" communications are needed to convince the public to use the Gardasil-9 vaccine, not necessarily that the communications need to be more "accurate". This seems to be the wrong adjective or descriptive word to use (to describe the inadequate public communications messaging).

A4. “Effective” was indeed a better word than “accurate”, and we corrected accordingly.

Round 2

Reviewer 2 Report

The manuscript is significantly improved and still needs only some minor suggested revisions to correct for more effective communication.

Title: remove space after hyphen  (i.e. Real-life) no space before "life"; also please revise the ending of the title as follows:     

.......profile of the 9-valent HPV vaccine based on data from the Puglia Region of southern Italy  [remove words "from routine surveillance"]

Materials and Methods

The first sentence needs to be combined with the following paragraph.

This is a retrospective observational study. Data were collected from....

Results

The first sentence should be a direct statement (not beginning with the preposition "from") as follows:

A total of 266,647 doses of 9vHPVv (Gardasil-9®) were administered in Puglia from January 2018 through November 2021.

L 178 [a 1-sentence paragraph] Add another descriptive sentence (following) to explain the information to be presented in the following 5 paragraphs [AEFI 1-5]

Author Response

Q1. The manuscript is significantly improved and still needs only some minor suggested revisions to correct for more effective communication.

A1. Thank you. We made all requested changes

Q2. Title: remove space after hyphen  (i.e. Real-life) no space before "life"; also please revise the ending of the title as follows:    

.......profile of the 9-valent HPV vaccine based on data from the Puglia Region of southern Italy  [remove words "from routine surveillance"]

A2. Ok.

Q3. Materials and Methods

The first sentence needs to be combined with the following paragraph.

This is a retrospective observational study. Data were collected from....

A3. Ok.

Q4. Results

The first sentence should be a direct statement (not beginning with the preposition "from") as follows:

A total of 266,647 doses of 9vHPVv (Gardasil-9®) were administered in Puglia from January 2018 through November 2021.

A4. Ok

Q5. L 178 [a 1-sentence paragraph] Add another descriptive sentence (following) to explain the information to be presented in the following 5 paragraphs [AEFI 1-5]

A5. Ok.